# Towards a Pairwise Ranking Model with Orderliness and Monotonicity for Label Enhancement

**Yunan Lu**[1,2], **Xixi Zhang**[1], **Yaojin Lin**[3], **Weiwei Li**[4], **Lei Yang**[2], **Xiuyi Jia**[1]*

[1] Nanjing University of Science and Technology, Nanjing, China
[2] The Hong Kong Polytechnic University, Hong Kong, China
[3] Minnan Normal University, Fujian, China
[4] Nanjing University of Aeronautics and Astronautics, Nanjing, China
{luyn,zhangxixi,jiaxy}@njust.edu.cn, yjlin@mnnu.edu.cn,
liweiwei@nuaa.edu.cn, ray.yang@polyu.edu.hk

## Abstract

Label distribution in recent years has been applied in a diverse array of complex decision-making tasks. To address the availability of label distributions, label enhancement has been established as an effective learning paradigm that aims to automatically infer label distributions from readily available multi-label data, e.g., logical labels. Recently, numerous works have demonstrated that the label ranking is significantly beneficial to label enhancement. However, these works still exhibit deficiencies in representing the probabilistic relationships between label distribution and label rankings, or fail to accommodate scenarios where multiple labels are equally important for a given instance. Therefore, we propose PROM, a pairwise ranking model with orderliness and monotonicity, to explain the probabilistic relationship between label distributions and label rankings. Specifically, we propose the monotonicity and orderliness assumptions for the probabilities of different ranking relationships and derive the mass functions for PROM, which are theoretically ensured to preserve the monotonicity and orderliness. Further, we propose a generative label enhancement algorithm based on PROM, which directly learns a label distribution predictor from the readily available multi-label data. Finally, extensive experiments demonstrate the efficacy of our proposed model.

## 1 Introduction

Label polysemy, i.e., the cases where an instance is associated with multiple labels simultaneously, is a common phenomenon in real-world tasks. To preserve label polysemy, multi-label learning (Tsoumakas and Katakis, 2006) employs binary values to indicate the presence or absence of each label for a given instance. However, multi-label learning cannot directly handle a further question with more polysemy: How much is each label associated with an instance? Hence, Geng (2016) introduced LDL (Label Distribution Learning). Unlike multi-label learning, LDL assigns a real-valued vector to each instance, resembling probability distributions, where each element, called label description degree, represents the extent to which the label describes the instance. By providing richer label information, LDL has been applied in various fields, including age estimation (Gao et al., 2018, Geng et al., 2013) and affective analysis (Jia et al., 2019, Machajdik and Hanbury, 2010).

A significant bottleneck hindering the broader application of LDL is the challenge of acquiring ground-truth label distributions, as accurately quantifying these distributions can be costly. To address

---

*Corresponding Author

39th Conference on Neural Information Processing Systems (NeurIPS 2025).

this issue, LE (Label Enhancement) has been proposed to automatically infer label distributions from more readily available multi-label data by mining the underlying label polysemy information (Xu et al., 2018, 2020). Initially, most LE methods mainly focused on mining the underlying label polysemy information from instance correlation and label correlation. Recently, some novel findings (Jia et al., 2023b, 2024, Wang and Geng, 2021) that the label ranking is beneficial for improving the generalization of LDL have inspired a portion of LE research to explore the label polysemy information from the perspective of label ranking. For example, Jia et al. (2023a), Lu et al. (2023a,b) proposed to incorporate the ranking relationship between positive and negative labels into the loss function or the generation model of label distributions. Although these works have made headway in designing methods for regularizing LE processes by label rankings, there remain the following two research gaps. 1) Current works ignore the ranking relation of tie, i.e. the cases where multiple labels simultaneously describe the instance to almost the same degree, which is ubiquitous in real-world tasks. 2) Current works neglect the qualitative and quantitative modeling of the probabilistic relationships between label distributions and label rankings, which is essential for the interpretable and generative modeling of label enhancement. Besides, it should be noted that while some classic ranking models, such as Bradley-Terry (Hunter, 2004) and Plackett-Luce (Guiver and Snelson, 2009) models, can be employed as a quantitative model, they also cannot explicitly model the tie relation.

Therefore, we propose PROM (a Pairwise Ranking model with Orderliness and Monotonicity) to qualitatively and quantitatively explain the probabilistic relationship between label distributions and label rankings, which can serve as a generative distribution of label rankings in generative label enhancement or as a loss function for measuring the inconsistency between label rankings and label distributions. Specifically, we first conduct a qualitative analysis of the probabilistic principles governing the generation process of label rankings from label distributions, and formalize these principles as the assumptions on the monotonicity and orderliness of the probabilities of different ranking relationships. Second, we design parameterized probability mass functions for the ranking model, and derive the conditions under which the model adheres to probabilistic monotonicity and orderliness. Third, we propose a generative label enhancement algorithm based on the PROM model, called LE-PROM, which integrates the LE process and the LDL process into a unified framework, and directly learns an LDL mapping on the training instances with multi-label data. Finally, we validate our proposal through extensive experiments on real-world datasets. The experimental results demonstrate the superiority of our proposed method.

## 2   Related Work

To tackle the challenge of acquiring accurate ground-truth label distributions, LE (Label Enhancement) was proposed to automatically infer label distributions from more readily available multi-label data, such as logical labels (Xu et al., 2018, Kou et al., 2025), ternary labels Lu and Jia (2024), multi-label rankings (Lu and Jia, 2022, Lu et al., 2023c), or inaccurate label distributions Kou et al. (2024, 2023), Lu et al. (2025). Most LE methods mine the label polysemy from instance and label correlations. For example, in order to capture the instance correlation, the prototype-based LE algorithms (El Gayar et al., 2006, Jiang et al., 2006, Wang et al., 2023, Fan et al., 2024) identify the representative points and subsequently estimate the label distribution based on the representative points of each label. The graph-based LE algorithms (Xu et al., 2018, Zhang et al., 2021, Xu et al., 2019, Liu et al., 2021) directly calculate affinities between instances based on the features, which is further utilized to regularize the label distribution matrix. The manifold-based LE algorithms (Hou et al., 2016, Wen et al., 2021, Tang et al., 2020, Zhang et al., 2018) learn coefficients for each instance, by which the feature vector of each instance can be linearly reconstructed from its neighbors, and then maintain the reconstruction within label distributions. In order to capture the label correlation, Luo et al. (2021) utilized the confusion matrix to estimate global label correlation. Besides, several works address the LE task under special data distribution. For example, VIB-ILE Song et al. (2024) addresses imbalanced label information in LE by adopting a variational information bottleneck mechanism and introducing consistency regularization.

Recently, a portion of LE research attempts to explore the label polysemy information from the perspective of label ranking. For example, Jia et al. (2023a) utilized a margin-based loss function to penalize the cases where the description degree of negative labels exceeds that of positive labels; Lu et al. (2023a) designs a conditional distribution of logical labels given label distributions, whose

parameters are regularized by the ranking between negative and positive labels. Lu et al. (2023b) derives a variational approximation for label distribution in generative label enhancement, which is theoretically assured to uphold the ranking relation between negative and positive labels. Besides, Lu and Jia (2022), Lu et al. (2023c) theoretically and methodologically investigated the task of predicting label distributions directly from multi-label rankings. Despite the success of these efforts, there remains a gap in the qualitative and quantitative modeling of the probabilistic relationships between label distributions and the tie-allowed label rankings.

# 3  Methodology

In this section, we first introduce the notations commonly used in this paper, secondly elaborate our proposed PROM model, and finally illustrate the generative LE framework.

## 3.1  Notations

We denote the $D$-dimensional feature space by $\mathcal{X}^D = \mathbb{R}^D$, denote the $M$-dimensional label distribution space by $\Delta^M = \{\boldsymbol{v} \in \mathbb{R}_+^M : \sum_{m=1}^M v_m = 1\}$, denote the label space by $\mathcal{Y} = \{\ell_m\}_{m=1}^M$. We cope with the training datasets that appear as data pairs $\{(\boldsymbol{x}_n, \boldsymbol{y}_n)\}_{n=1}^N$, where $\boldsymbol{x}_n \in \mathcal{X}^D$ and $\boldsymbol{y}_n$ denote the feature vector and the vector of the easily available label values of the $n$-th instance, respectively. The data of $\boldsymbol{y}_n$ can directly yield pairwise ranking relation among labels. We denote the ranking relation between label $\ell_i$ and label $\ell_j$ by $\xi_{ij} \in \{\ell_i \prec \ell_j, \ell_i \succ \ell_j, \ell_i \simeq \ell_j\}$, where $\ell_i \prec \ell_j$, $\ell_i \succ \ell_j$, and $\ell_i \simeq \ell_j$ denote that the label $\ell_i$, compared to the label $\ell_j$, describes the instance to a higher, lower, and approximately the same degree, respectively. The goal of LE is to learn a label distribution predictor $f : \mathcal{X}^D \to \Delta^M$, i.e., a mapping from feature space to label distribution space, by the training dataset $\{(\boldsymbol{x}_n, \boldsymbol{y}_n)\}_{n=1}^N$.

## 3.2  PROM

In this subsection, we first propose assumptions to formalize the monotonicity and orderliness of ranking probability. Then, we derive the parametric form of PROM.

### 3.2.1  Fundamental Assumptions of PROM

On the one hand, we study how the label description degree affects the probability of the label ranking, focusing on the monotonicity of the probability of the label ranking. Obviously, given any two labels $\ell_i, \ell_j$ of an instance, the label $\ell_i$ is more likely to rank above the label $\ell_j$ (or the label $\ell_j$ is more likely to rank below the label $\ell_i$ alternatively) if the description degree of $\ell_i$ exceeds that of $\ell_j$ by a larger margin. Besides, if the description degrees of two labels are close, then the strict ranking relation between the two labels will be difficult to distinguish, i.e., it is more likely that the two labels are tied. We formalize the above intuition as the probability monotonicity assumption.

**Assumption 3.1** (Probability monotonicity)**.** Given any two labels $\ell_i, \ell_j$ of an instance and their respective description degrees $z_i, z_j$, $p(\ell_i \prec \ell_j | z_i = u_i, z_j = u_j) < p(\ell_i \prec \ell_j | z_i = v_i, z_j = v_j)$ and $p(\ell_i \succ \ell_j | z_i = u_i, z_j = u_j) > p(\ell_i \succ \ell_j | z_i = v_i, z_j = v_j)$ holds for any $u_i - u_j < v_i - v_j$. $p(\ell_i \simeq \ell_j | z_i = u_i, z_j = u_j) > p(\ell_i \simeq \ell_j | z_i = v_i, z_j = v_j)$ holds for any $|u_i - u_j| < |v_i - v_j|$, where $u_i, u_j, v_i, v_j \in [0, 1]$.

On the other hand, we explore how the label description degree affects the orderliness among the probabilities of different rankings. Obviously, given any two labels $\ell_i, \ell_j$ of an instance, these two labels are most likely to be tied if the description degrees of the two labels are sufficiently close; the label $\ell_i$ is most likely to rank above the label $\ell_j$ (or the label $\ell_j$ is most likely to rank below the label $\ell_i$ alternatively) if the label description degree of $\ell_i$ exceeds that of $\ell_j$ by a sufficiently large margin. We formalize the above intuition as the probability orderliness assumption.

**Assumption 3.2** (Probability orderliness)**.** There exists two real thresholds $-1 < z_- < 0 < z_+ < 1$, $p(\ell_i \succ \ell_j | z_i, z_j) > p(\ell_i \simeq \ell_j | z_i, z_j) > p(\ell_i \prec \ell_j | z_i, z_j)$ holds for any $z_i - z_j < z_-$; $p(\ell_i \succ \ell_j | z_i, z_j) < p(\ell_i \simeq \ell_j | z_i, z_j) < p(\ell_i \prec \ell_j | z_i, z_j)$ holds for any $z_i - z_j > z_+$; $p(\ell_i \simeq \ell_j | z_i, z_j) > \max\{p(\ell_i \succ \ell_j | z_i, z_j), p(\ell_i \prec \ell_j | z_i, z_j)\}$ holds for any $z_- < z_i - z_j < z_+$.

The intuition of probability monotonicity and orderliness is visualized in Figure 1. Actually, the above discussion also implicitly adheres to the following two assumptions. The one is that the ranking relation between two labels depends solely on the difference in their description degrees, and is independent of the description degrees themselves. We refer to this assumption as the translational invariance assumption. The other is that the ranking relation between two labels is symmetric, i.e., $\ell_i \prec \ell_j$ is actually $\ell_j \succ \ell_i$. We refer to this assumption as the ranking symmetry assumption. We formalize these two assumptions as follows.

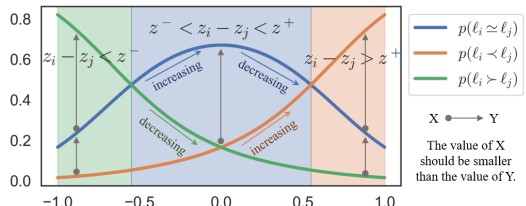

Figure 1: Schematic of probability monotonicity and probability orderliness.

**Assumption 3.3** (Translational invariance). Given any two labels $\ell_i, \ell_j$ of an instance and their respective description degrees $z_i, z_j$, $p(\ell_i \prec \ell_j | z_i = u_i, z_j = u_j) = p(\ell_i \prec \ell_j | z_i = v_i, z_j = v_j)$ and $p(\ell_i \succ r_j | z_i = u_i, z_j = u_j) = p(\ell_i \succ \ell_j | z_i = v_i, z_j = v_j)$ hold for any $u_i - u_j = v_i - v_j$; $p(\ell_i \simeq \ell_j | z_i = u_i, z_j = u_j) = p(\ell_i \simeq \ell_j | z_i = v_i, z_j = v_j)$ holds for any $|u_i - u_j| = |v_i - v_j|$, where $u_i, u_j, v_i, v_j \in [0, 1]$.

**Assumption 3.4** (Ranking symmetry). Given any two labels $\ell_i, \ell_j$ of an instance and their respective description degrees $z_i, z_j$, $p(\ell_i \prec \ell_j | z_i, z_j) = p(\ell_j \succ \ell_i | z_j, z_i)$ and $p(\ell_i \simeq \ell_j | z_i, z_j) = p(\ell_j \simeq \ell_i | z_j, z_i)$.

### 3.2.2 Probability Mass Functions of PROM

Since the ranking relation between two labels can be $\ell_i \prec \ell_j$, $\ell_i \succ \ell_j$, or $\ell_i \simeq \ell_j$, we model the ranking relation by a categorical distribution, which can be formalized as:

$$\xi_{ij} \mid z_i, z_j \sim \text{Categorical}(\xi_{ij} \mid [\underline{\phi}(z_i, z_j), \tilde{\phi}(z_i, z_j), \overline{\phi}(z_i, z_j)]), \tag{1}$$

where $\underline{\phi}(z_i, z_j)$, $\overline{\phi}(z_i, z_j)$, and $\tilde{\phi}(z_i, z_j)$ capture the rules of generating the label ranking relation $\ell_i \succ \ell_j$, $\ell_i \prec \ell_j$, and $\ell_i \simeq \ell_j$, respectively. Considering that the softmax function is most commonly used to model the probability of discrete events, we preliminarily assume the parametric form of $\phi(z_i, z_j)$ as follows:

$$\begin{aligned}
\underline{\phi}(z_i, z_j) &= p(\ell_i \succ \ell_j | z_i, z_j) = Z^{-1} \exp(a_{\underline{\phi}}(z_i - z_j) + b_{\underline{\phi}}), \\
\tilde{\phi}(z_i, z_j) &= p(\ell_i \simeq \ell_j | z_i, z_j) = Z^{-1} \exp(a_{\tilde{\phi}}(z_i - z_j)^2 + b_{\tilde{\phi}}), \\
\overline{\phi}(z_i, z_j) &= p(\ell_i \prec \ell_j | z_i, z_j) = Z^{-1} \exp(a_{\overline{\phi}}(z_i - z_j) + b_{\overline{\phi}}),
\end{aligned} \tag{2}$$

where $Z$ is a normalization factor that ensures $\underline{\phi}(z_i, z_j) + \tilde{\phi}(z_i, z_j) + \overline{\phi}(z_i, z_j) = 1$, $a_{\underline{\phi}} < 0$, $a_{\tilde{\phi}} < 0$, $a_{\overline{\phi}} > 0$, and $b_{\underline{\phi}}$, $b_{\tilde{\phi}}$, and $b_{\overline{\phi}}$ are arbitrary real numbers.

Figure 2 visualizes PROM model at different values of parameters. It is evident that not any value of parameter can adhere to the probability monotonicity assumption and the probability orderliness assumption. For example, Figure 2(a) violates the probability monotonicity assumption, Figure 2(b) violates the probability orderliness assumption, and Figure 2(c) violates both. Figure 2(d) shows a PROM model that we expect. Therefore, we next propose four theorems to ensure that the probability mass functions defined by Equation (2) adhere to the proposed assumptions.

**Theorem 3.5.** $p(\xi_{ij} | z_i, z_j)$ *defined by Equation* (1) *and Equation* (2) *adheres to the translation invariance assumption, i.e. Assumption 3.3.*

**Theorem 3.6.** $p(\xi_{ij} | z_i, z_j)$ *defined by Equation* (1) *and Equation* (2) *adheres to the ranking symmetry assumption, i.e. Assumption 3.4, iff* $a_{\overline{\phi}} = -a_{\underline{\phi}}$ *and* $b_{\overline{\phi}} = b_{\underline{\phi}}$.

**Theorem 3.7.** *Given* $a_{\overline{\phi}} > 0$, $a_{\underline{\phi}} < 0$, $a_{\tilde{\phi}} < 0$, $\Delta = z_i - z_j$. $\tilde{\phi}$ *adheres to the monotonicity assumption of* $p(\ell_i \simeq \ell_j | z_i, z_j)$ *if* $a_{\overline{\phi}} = -a_{\underline{\phi}} \exp(b_{\underline{\phi}} - b_{\overline{\phi}})$; $\overline{\phi}$ *adheres to the monotonicity assumption of* $p(\ell_i \prec \ell_j | z_i, z_j)$ *if* $(a_{\overline{\phi}}(2a_{\tilde{\phi}})^{-1} < -1) \vee (\overline{\Delta}^\star \leq -1 \wedge h_{\overline{\phi}}(-1) > 0) \vee (\overline{\Delta}^\star > -1 \wedge h_{\overline{\phi}}(\overline{\Delta}^\star) > 0)$; $\underline{\phi}$ *adheres to the monotonicity assumption of* $p(\ell_i \succ \ell_j | z_i, z_j)$

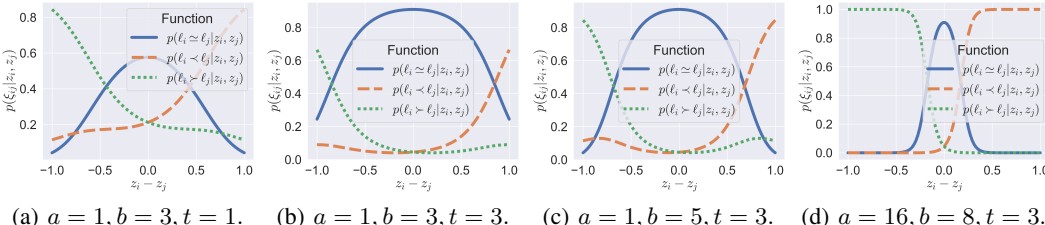

(a) $a = 1, b = 3, t = 1$.    (b) $a = 1, b = 3, t = 3$.    (c) $a = 1, b = 5, t = 3$.    (d) $a = 16, b = 8, t = 3$.

Figure 2: The shape of PROM with varying values of parameters.

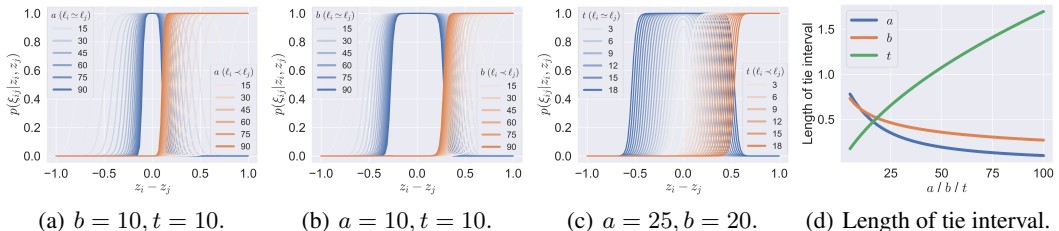

(a) $b = 10, t = 10$.    (b) $a = 10, t = 10$.    (c) $a = 25, b = 20$.    (d) Length of tie interval.

Figure 3: Marginal effect of parameters on the shape of PROM.

if $(a_{\underline{\phi}}(2a_{\tilde{\phi}})^{-1} > 1) \vee (\underline{\Delta}^\star \geq 1 \wedge h_{\underline{\phi}}(1) > 0) \vee (\underline{\Delta}^\star < 1 \wedge h_{\underline{\phi}}(\underline{\Delta}^\star) > 0)$ where $\overline{\Delta}^\star$, $\underline{\Delta}^\star$, $h_{\overline{\phi}}(\Delta)$, and $h_{\underline{\phi}}(\Delta)$ are defined by $\overline{\Delta}^\star := (a_{\overline{\phi}} + a_{\underline{\phi}} + \sqrt{(a_{\overline{\phi}} - a_{\underline{\phi}})^2 - 8a_{\tilde{\phi}}})(4a_{\tilde{\phi}})^{-1}$, $\underline{\Delta}^\star := (a_{\overline{\phi}} + a_{\underline{\phi}} - \sqrt{(a_{\overline{\phi}} - a_{\underline{\phi}})^2 - 8a_{\tilde{\phi}}})(4a_{\tilde{\phi}})^{-1}$, $h_{\overline{\phi}}(\Delta) := a_{\underline{\phi}}\Delta - a_{\tilde{\phi}}\Delta^2 + b_{\underline{\phi}} - b_{\tilde{\phi}} + \log \frac{a_{\overline{\phi}} - a_{\underline{\phi}}}{2a_{\tilde{\phi}}\Delta - a_{\overline{\phi}}}$, $h_{\underline{\phi}}(\Delta) := -a_{\tilde{\phi}}\Delta^2 + a_{\overline{\phi}}\Delta + b_{\overline{\phi}} - b_{\tilde{\phi}} + \log \frac{a_{\overline{\phi}} - a_{\underline{\phi}}}{a_{\underline{\phi}} - 2a_{\tilde{\phi}}\Delta}$.

**Theorem 3.8.** $p(\xi_{ij}|z_i, z_j)$ defined by Equation (1) and Equation (2) adheres to the probability orderliness assumption, i.e. Assumption 3.2, iff $\max\{b_{\overline{\phi}} - a_{\overline{\phi}}, b_{\underline{\phi}} + a_{\underline{\phi}}\} < a_{\tilde{\phi}} + b_{\tilde{\phi}} < \min\{b_{\overline{\phi}} + a_{\overline{\phi}}, b_{\underline{\phi}} - a_{\underline{\phi}}\}$ and $b_{\tilde{\phi}} > \max\{b_{\overline{\phi}}, b_{\underline{\phi}}\}$.

The proof of the above theorems can be found in Appendix A. According to the above theorems, let $a = a_{\overline{\phi}} = -a_{\underline{\phi}}$, $b = -a_{\tilde{\phi}}$, $b_{\overline{\phi}} = b_{\underline{\phi}}$, $t = b_{\tilde{\phi}} - b_{\overline{\phi}}$, we finally define the probability mass function of PROM model as follows:

$$\underline{\phi}(z_i, z_j) = Z^{-1}e^{-a(z_i - z_j)}, \quad \tilde{\phi}(z_i, z_j) = Z^{-1}e^{-b(z_i - z_j)^2 + t}, \quad \overline{\phi}(z_i, z_j) = Z^{-1}e^{a(z_i - z_j)}, \quad (3)$$

where the parameters are bounded by $(a, b, t) \in \mathcal{F}$, where the feasible region $\mathcal{F}$ is defined by Equation (4):

$$\mathcal{F} = \Big\{(a, b, t) \mid (a > |b - t| \wedge b > 0 \wedge t > 0) \wedge \Big( (a^2 \geq 4b^2 - 2b \wedge t < b + a - \log \frac{2b - a}{2a}) \\ \vee (a \geq 2b) \vee \big( t < \frac{2a\sqrt{a^2 + 2b} + a^2}{4b} - \log \frac{\sqrt{a^2 + 2b} - a}{2a} + \frac{1}{2} \wedge a^2 < 4b^2 - 2b \big) \Big) \Big\}. \quad (4)$$

### 3.2.3 Marginal Effect of Parameters on PROM

Intuitively, the parameters $a$ and $b$ control the uncertainty of $\ell_i \prec \ell_j$ (or $\ell_i \succ \ell_j$) and $\ell_i \simeq \ell_j$, respectively. As shown in Figure 3(a) and Figure 3(b), larger $a$ and $b$ correspond to the steeper density curve of the probabilities of $\ell_i \prec \ell_j$ (or $\ell_i \succ \ell_j$) and $\ell_i \simeq \ell_j$, respectively. More results are shown in Appendix B. We do not show $p(\ell_i \succ \ell_j|z_i, z_j)$ for the sake of readability. The parameter $t$ is similar to the temperature coefficient in the softmax function, whose effect on the PROM distribution is shown in Figure 3(c). Overall, these three hyper-parameters directly affect the length of tie interval of PROM distribution, i.e., the length of the interval of $z_i - z_j$ with $p(\ell_i \simeq \ell_j) > \max\{p(\ell_i \prec \ell_j), p(\ell_i \succ \ell_j)\}$, which is shown in Figure 3(d).

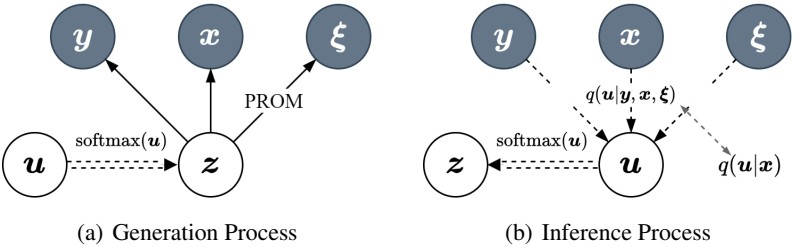

(a) Generation Process  (b) Inference Process

Figure 4: Schematic diagram of LE-PROM.

## 3.3 LE-PROM

LE-PROM consists of a probabilistic generation process for observed variables (features, available labels, pairwise rankings) and a posterior inference process for latent variables (label distribution), which will be illustrated in the following two subsections.

### 3.3.1 Generation Process

Here we illustrate the process of generating the observations, which is shown in Figure 4(a) and formalized as follows.

1. Generate a sample of label distribution:
   (a) Generate a real-valued vector $\boldsymbol{u}$ from a standard multivariate normal distribution, i.e., $\boldsymbol{u} \sim \mathcal{N}(\boldsymbol{u} \mid \boldsymbol{0}_M, \boldsymbol{I}_M)$, where $\boldsymbol{0}_M$ is an $M$-dimensional all-zero vector, $\boldsymbol{I}_M$ is an $M \times M$ identity matrix.
   (b) Transform the real-valued vector $\boldsymbol{u}$ into the label distribution $\boldsymbol{z}$, i.e., $\boldsymbol{z} = \text{softmax}(\boldsymbol{u})$.
2. Generate the available label $\boldsymbol{y}$ from a task-specific distribution given $\boldsymbol{z}$: $\boldsymbol{y} \mid \boldsymbol{z} \sim \tilde{p}(\boldsymbol{y}|\boldsymbol{z})^2$.
3. Generate a sample of feature variables $\boldsymbol{x}$ from a multivariate normal distribution conditioned on the label distribution[3]. Formally, $\boldsymbol{x} \mid \boldsymbol{z} \sim \mathcal{N}(\boldsymbol{x} \mid \mu_{\boldsymbol{x}}(\boldsymbol{z}), \text{diag}(\sigma_{\boldsymbol{x}}^2(\boldsymbol{z})))$, where $\mu_{\boldsymbol{x}}(\cdot)$ and $\sigma_{\boldsymbol{x}}(\cdot)$ are two neural networks, and $\text{diag}(\sigma)$ is a diagonal matrix whose $(i, i)$ entry is the $i$-th element of $\sigma$.
4. Generate a sample of pairwise rankings $\boldsymbol{\xi}$ from our designed PROM distribution conditioned on the label distribution $\boldsymbol{z}$. Formally, $\boldsymbol{\xi} \mid \boldsymbol{z} \sim \prod_{i<j} \text{PROM}(\xi_{ij} \mid z_i, z_j)$.

According to the above generation process, the joint distribution of the complete-data can be factorized as $p(\boldsymbol{u}, \boldsymbol{x}, \boldsymbol{y}, \boldsymbol{\xi}) = p(\boldsymbol{u})p(\boldsymbol{x}|\boldsymbol{u})p(\boldsymbol{y}|\boldsymbol{u})p(\boldsymbol{\xi}|\boldsymbol{u})$. It should be noted that we omit label distribution $\boldsymbol{z}$ since $\boldsymbol{z}$ can be deterministically obtained from $\boldsymbol{u}$.

### 3.3.2 Model Inference

Here we derive the inference method of our proposed generative model. The schematic is shown in Figure 4(b). We employ variational inference to approximate the true posterior, since the exact inference of the posterior of $\boldsymbol{z}$ is intractable due to the nonlinear and non-conjugate relationships among variables. First, we assume that the variational posterior of $\boldsymbol{u}$ can be decomposed as

$$q(\boldsymbol{u}|\boldsymbol{x}, \boldsymbol{y}, \boldsymbol{\xi}) = \mathcal{N}(\boldsymbol{u} \mid \mu_{\boldsymbol{u}}(\boldsymbol{x}, \boldsymbol{y}, \boldsymbol{\xi}), \text{diag}(\sigma_{\boldsymbol{u}}(\boldsymbol{x}, \boldsymbol{y}, \boldsymbol{\xi}))). \tag{5}$$

Then, we find the variational posterior distribution of $\boldsymbol{u}$ that is closest to the true posterior distribution. It has been proven that the Kullback-Leibler (KL) divergence between the variational posterior and the true posterior can be minimized by maximizing the ELBO (evidence lower bound):

$$\text{ELBO} = \mathbb{E}_{q(\boldsymbol{u}|\boldsymbol{x}, \boldsymbol{y}, \boldsymbol{\xi})}[\log p(\boldsymbol{x}, \boldsymbol{y}, \boldsymbol{\xi}|\boldsymbol{u})] - \mathcal{D}_{\text{KL}}(q(\boldsymbol{u}|\boldsymbol{x}, \boldsymbol{y}, \boldsymbol{\xi})\|p(\boldsymbol{u})). \tag{6}$$

In the right-hand side of Equation (6), the first term is also known as the reconstruction term, and the second term is also known as the prior regularization term. The reconstruction term serves to

---

[2]The task-specific distribution can be conditioned on $\boldsymbol{u}$ or $\boldsymbol{z}$ since $\boldsymbol{z}$ can be deterministically obtained by $\boldsymbol{u}$.

[3]Analogous to $\tilde{p}(\boldsymbol{y}|\boldsymbol{z})$, the multivariate normal distribution for features can also be conditioned on $\boldsymbol{z}$ or $\boldsymbol{u}$.

ensure that the label distribution can accurately reconstruct the observed variables. It is evident that an analytic form of reconstruction term is intractable due to the integral of a complicated likelihood function. Therefore, we turn to SGVB estimator. Since $q(\boldsymbol{u}|\boldsymbol{x},\boldsymbol{y},\boldsymbol{\xi})$ is multivariate normal, it can be reparameterized by $q(\boldsymbol{u}|\boldsymbol{x},\boldsymbol{y},\boldsymbol{\xi}) = \mu_{\boldsymbol{u}}(\boldsymbol{x},\boldsymbol{y},\boldsymbol{\xi}) + \sigma_{\boldsymbol{u}}(\boldsymbol{x},\boldsymbol{y},\boldsymbol{\xi}) \odot \boldsymbol{\epsilon}$, where $\boldsymbol{\epsilon}$ is standard normal noise. Besides, benefiting from (Lu et al., 2023a), we utilize hyperparameters $\alpha$ and $\beta$ to re-weight the reconstruction quality of $\boldsymbol{x}$, $\boldsymbol{y}$, and $\boldsymbol{u}$ to re-balance the magnitudes of different likelihood functions. Finally, the reconstruction term can be estimated by:

$$\mathbb{E}_{q(\boldsymbol{u}|\boldsymbol{x},\boldsymbol{y},\boldsymbol{\xi})}[\log p(\boldsymbol{x},\boldsymbol{y},\boldsymbol{\xi}|\boldsymbol{u})] \approx \frac{1}{(1+\alpha+\beta)L} \sum_{t=1}^{L} \log p\left(\boldsymbol{x}|\mu_{\boldsymbol{u}}(\boldsymbol{x},\boldsymbol{y},\boldsymbol{\xi}) + \sigma_{\boldsymbol{u}}(\boldsymbol{x},\boldsymbol{y},\boldsymbol{\xi}) \odot \boldsymbol{\epsilon}^{(t)}\right)$$

$$+\alpha \log p\left(\boldsymbol{y}|\mu_{\boldsymbol{u}}(\boldsymbol{x},\boldsymbol{y},\boldsymbol{\xi}) + \sigma_{\boldsymbol{u}}(\boldsymbol{x},\boldsymbol{y},\boldsymbol{\xi}) \odot \boldsymbol{\epsilon}^{(t)}\right) + \beta \log p\left(\boldsymbol{\xi}|\mu_{\boldsymbol{u}}(\boldsymbol{x},\boldsymbol{y},\boldsymbol{\xi}) + \sigma_{\boldsymbol{u}}(\boldsymbol{x},\boldsymbol{y},\boldsymbol{\xi}) \odot \boldsymbol{\epsilon}^{(t)}\right),$$

$$(7)$$

where $L$ is the number of Monte Carlo samples, $\boldsymbol{\epsilon}^{(t)}$ is a sample from the standard normal distribution. The prior regularization term encourages the posterior to approach the prior. Since both the $\boldsymbol{u}$-posterior and $\boldsymbol{u}$-prior are multivariate normal, their KL divergence can be computed analytically. Besides, in order to train a feature-conditioned label distribution predictor $\hat{p}(\boldsymbol{u}|\boldsymbol{x}) = \mathcal{N}(\boldsymbol{u} \mid \mu(\boldsymbol{x}), \mathrm{diag}(\sigma(\boldsymbol{x})))$ during the model inference phase, we add a prediction loss term to ELBO. Finally, the optimization objective can be formalized as follows:

$$\arg\max \mathbb{E}_{q(\boldsymbol{u}|\boldsymbol{x},\boldsymbol{y},\boldsymbol{\xi})}[\log p(\boldsymbol{x},\boldsymbol{y},\boldsymbol{\xi}|\boldsymbol{u})] - \mathcal{D}_{\mathrm{KL}}(q(\boldsymbol{u}|\boldsymbol{x},\boldsymbol{y},\boldsymbol{\xi})\|p(\boldsymbol{u})) - \lambda\mathcal{D}_{\mathrm{KL}}(\hat{p}(\boldsymbol{u}|\boldsymbol{x})\|q(\boldsymbol{u}|\boldsymbol{x},\boldsymbol{y},\boldsymbol{\xi})),$$

$$(8)$$

where $\lambda$ is a trade-off hyperparameter. As suggested by Daunizeau (2017), since $\boldsymbol{z}$ is the softmax normalization of a Gaussian random vector, the expected label distribution based on $\hat{p}(\boldsymbol{u}|\boldsymbol{x})$ can be approximated as follows:

$$\mathbb{E}_{\hat{p}(\boldsymbol{u}|\boldsymbol{x})}[z_m] \approx \frac{1}{\sum_{i=1}^{M} \exp(\psi_{mi})}, \quad \psi_{mi} = \frac{\mu_i - \mu_m}{\sqrt{1 + 3\pi^{-2}(\sigma_m^2 + \sigma_i^2)}}, \tag{9}$$

where $\mu_i$ and $\sigma_i$ denote the $i$-th elements of $\mu(\boldsymbol{x})$ and $\sigma(\boldsymbol{x})$, respectively.

# 4 Experiments

## 4.1 Datasets and Evaluation Measures

We conduct experiments on 16 real-world datasets. These datasets are collected from diverse real-world tasks, including emotion analysis, movie rating prediction, and bioinformatics. Specifically, "Painting" (Machajdik and Hanbury, 2010), "Emotion6" (Peng et al., 2015), "Music" (Lee et al., 2021), "BU-3DFE" (Yin et al., 2006), and "JAFFE" (Lyons et al., 1998) are collected from emotion analysis tasks. "Movie" (Geng, 2016) is a dataset from movie rating prediction task. "Alpha", "Cdc", "Cold", "Diau", "Dtt", "Elu", "Heat", "Spo", "Spo5", and "Spoem" (Geng, 2016) are collected from a gene expression analysis task. To accelerate convergence, we use min-max normalization to preprocess the feature data. We evaluate the model performance by two representative LDL measures: KL (Kullback-Leibler divergence) and Cosine (cosine similarity). The KL (or Cosine) metric with smaller values represents the better (or worse) performance. More details of the datasets and the results on more metrics can be found in Appendix C.

## 4.2 Experiments on Logical Label Enhancement

Here, we aim to experimentally answer the question of *whether our proposed LE-PROM is superior to the state-of-the-art logical label enhancement algorithms.*

### 4.2.1 Experimental Configurations

**Experimental Method.** Initially, we randomly divide the dataset, allocating 70% of the dataset to the training set and the remaining 30% to the testing set. Then we reduce the label distributions of the training instances to logical labels, and the detailed reduction algorithm can be found in Appendix C. Subsequently, we apply LE algorithms to transform the logical labels into the label distribution for each training instance, and utilize the recovered label distributions to train an LDL

| Dataset | LE-PROM | CWLD | VIB-ILE | GLLE | KMLE | FCMLE |
|---|---|---|---|---|---|---|
| | | | KL ($\downarrow$) | | | |
| Painting | (1) $0.567_{\pm0.022}$ | (5) $0.991_{\pm0.068}$● | (6) $27.118_{\pm2.496}$● | (2) $0.570_{\pm0.025}$● | (4) $0.978_{\pm0.217}$● | (3) $0.589_{\pm0.030}$● |
| Emotion6 | (1) $0.616_{\pm0.013}$ | (5) $0.686_{\pm0.014}$● | (6) $0.737_{\pm0.017}$● | (2) $0.627_{\pm0.013}$● | (3) $0.646_{\pm0.026}$● | (4) $0.682_{\pm0.012}$● |
| Movie | (1) $0.114_{\pm0.002}$ | (5) $0.211_{\pm0.016}$● | (4) $0.193_{\pm0.021}$● | (2) $0.115_{\pm0.003}$● | (6) $0.216_{\pm0.015}$● | (3) $0.178_{\pm0.003}$● |
| Music | (1) $0.113_{\pm0.007}$ | (4) $0.147_{\pm0.002}$● | (6) $0.225_{\pm0.012}$● | (2) $0.135_{\pm0.008}$● | (5) $0.187_{\pm0.094}$● | (3) $0.136_{\pm0.009}$● |
| BU-3DFE | (1) $0.072_{\pm0.001}$ | (5) $0.101_{\pm0.003}$● | (4) $0.096_{\pm0.003}$● | (1) $0.072_{\pm0.002}$ | (6) $0.111_{\pm0.009}$● | (3) $0.079_{\pm0.002}$● |
| JAFFE | (1) $0.054_{\pm0.005}$ | (5) $0.152_{\pm0.039}$● | (4) $0.084_{\pm0.006}$● | (2) $0.064_{\pm0.007}$● | (6) $0.409_{\pm0.168}$● | (3) $0.071_{\pm0.005}$● |
| Alpha | (1) $0.006_{\pm0.000}$ | (5) $0.012_{\pm0.001}$● | (1) $0.006_{\pm0.000}$ | (4) $0.010_{\pm0.001}$● | (6) $0.024_{\pm0.003}$● | (1) $0.006_{\pm0.000}$ |
| Cdc | (1) $0.007_{\pm0.000}$ | (5) $0.012_{\pm0.000}$● | (3) $0.008_{\pm0.000}$● | (4) $0.011_{\pm0.001}$● | (6) $0.019_{\pm0.001}$● | (1) $0.007_{\pm0.000}$ |
| Cold | (1) $0.012_{\pm0.001}$ | (5) $0.017_{\pm0.000}$● | (2) $0.013_{\pm0.001}$ | (4) $0.016_{\pm0.001}$● | (6) $0.027_{\pm0.002}$● | (2) $0.013_{\pm0.000}$● |
| Diau | (1) $0.014_{\pm0.001}$ | (3) $0.020_{\pm0.001}$● | (4) $0.022_{\pm0.001}$● | (4) $0.022_{\pm0.001}$● | (6) $0.075_{\pm0.003}$● | (2) $0.015_{\pm0.000}$● |
| Dtt | (2) $0.007_{\pm0.000}$ | (5) $0.010_{\pm0.000}$● | (1) $0.006_{\pm0.000}$○ | (4) $0.009_{\pm0.000}$● | (6) $0.014_{\pm0.002}$● | (2) $0.007_{\pm0.001}$ |
| Elu | (1) $0.006_{\pm0.000}$ | (5) $0.012_{\pm0.000}$● | (3) $0.007_{\pm0.001}$● | (4) $0.011_{\pm0.001}$● | (6) $0.019_{\pm0.001}$● | (1) $0.006_{\pm0.000}$ |
| Heat | (1) $0.013_{\pm0.000}$ | (5) $0.017_{\pm0.000}$● | (1) $0.013_{\pm0.000}$● | (4) $0.014_{\pm0.001}$● | (6) $0.018_{\pm0.001}$● | (1) $0.013_{\pm0.000}$ |
| Spo | (1) $0.025_{\pm0.001}$ | (5) $0.030_{\pm0.001}$● | (3) $0.027_{\pm0.001}$● | (2) $0.026_{\pm0.001}$● | (6) $0.034_{\pm0.002}$● | (3) $0.027_{\pm0.001}$● |
| Spo5 | (1) $0.030_{\pm0.001}$ | (6) $0.054_{\pm0.002}$● | (4) $0.034_{\pm0.001}$● | (3) $0.033_{\pm0.001}$● | (4) $0.034_{\pm0.001}$● | (1) $0.030_{\pm0.001}$○ |
| Spoem | (1) $0.026_{\pm0.001}$ | (6) $0.057_{\pm0.003}$● | (1) $0.026_{\pm0.001}$● | (1) $0.026_{\pm0.002}$● | (5) $0.029_{\pm0.005}$● | (4) $0.027_{\pm0.001}$● |
| | | | Cosine ($\uparrow$) | | | |
| Painting | (1) $0.718_{\pm0.009}$ | (4) $0.648_{\pm0.013}$● | (6) $0.281_{\pm0.076}$● | (1) $0.718_{\pm0.010}$ | (5) $0.633_{\pm0.025}$● | (3) $0.705_{\pm0.010}$● |
| Emotion6 | (1) $0.707_{\pm0.006}$ | (4) $0.668_{\pm0.007}$● | (4) $0.668_{\pm0.009}$● | (2) $0.701_{\pm0.005}$● | (3) $0.693_{\pm0.013}$● | (6) $0.665_{\pm0.004}$● |
| Movie | (1) $0.926_{\pm0.001}$ | (5) $0.892_{\pm0.003}$● | (3) $0.902_{\pm0.003}$● | (2) $0.925_{\pm0.002}$● | (4) $0.901_{\pm0.003}$● | (6) $0.871_{\pm0.002}$● |
| Music | (1) $0.915_{\pm0.005}$ | (4) $0.897_{\pm0.001}$● | (6) $0.832_{\pm0.008}$● | (2) $0.903_{\pm0.005}$● | (5) $0.877_{\pm0.041}$● | (3) $0.902_{\pm0.005}$● |
| BU-3DFE | (2) $0.929_{\pm0.001}$ | (5) $0.905_{\pm0.002}$● | (4) $0.910_{\pm0.002}$● | (1) $0.930_{\pm0.002}$ | (6) $0.902_{\pm0.002}$● | (3) $0.923_{\pm0.002}$● |
| JAFFE | (1) $0.948_{\pm0.006}$ | (5) $0.909_{\pm0.008}$● | (4) $0.920_{\pm0.006}$● | (2) $0.940_{\pm0.006}$● | (6) $0.816_{\pm0.047}$● | (3) $0.933_{\pm0.005}$● |
| Alpha | (1) $0.994_{\pm0.000}$ | (4) $0.990_{\pm0.000}$● | (1) $0.994_{\pm0.000}$ | (4) $0.990_{\pm0.001}$● | (6) $0.979_{\pm0.001}$● | (1) $0.994_{\pm0.001}$● |
| Cdc | (1) $0.993_{\pm0.000}$ | (5) $0.988_{\pm0.000}$● | (1) $0.993_{\pm0.001}$● | (4) $0.990_{\pm0.000}$● | (6) $0.981_{\pm0.001}$● | (1) $0.993_{\pm0.000}$ |
| Cold | (1) $0.988_{\pm0.000}$ | (5) $0.984_{\pm0.000}$● | (1) $0.988_{\pm0.000}$ | (4) $0.985_{\pm0.001}$● | (6) $0.973_{\pm0.002}$● | (1) $0.988_{\pm0.001}$● |
| Diau | (1) $0.987_{\pm0.000}$ | (3) $0.982_{\pm0.000}$● | (4) $0.980_{\pm0.001}$● | (5) $0.979_{\pm0.001}$● | (6) $0.939_{\pm0.002}$● | (2) $0.986_{\pm0.000}$● |
| Dtt | (1) $0.994_{\pm0.000}$ | (5) $0.990_{\pm0.000}$● | (1) $0.994_{\pm0.000}$ | (4) $0.992_{\pm0.001}$● | (6) $0.987_{\pm0.002}$● | (1) $0.994_{\pm0.000}$ |
| Elu | (1) $0.994_{\pm0.000}$ | (5) $0.988_{\pm0.000}$● | (1) $0.994_{\pm0.000}$● | (4) $0.990_{\pm0.000}$● | (6) $0.982_{\pm0.001}$● | (1) $0.994_{\pm0.000}$ |
| Heat | (1) $0.988_{\pm0.000}$ | (5) $0.983_{\pm0.000}$● | (1) $0.988_{\pm0.000}$ | (4) $0.986_{\pm0.001}$● | (6) $0.982_{\pm0.001}$● | (1) $0.988_{\pm0.000}$ |
| Spo | (1) $0.977_{\pm0.001}$ | (5) $0.972_{\pm0.001}$● | (2) $0.975_{\pm0.001}$● | (2) $0.975_{\pm0.001}$● | (6) $0.968_{\pm0.002}$● | (2) $0.975_{\pm0.001}$● |
| Spo5 | (2) $0.973_{\pm0.001}$ | (6) $0.956_{\pm0.001}$● | (4) $0.970_{\pm0.001}$● | (3) $0.971_{\pm0.001}$● | (4) $0.970_{\pm0.001}$● | (1) $0.974_{\pm0.000}$○ |
| Spoem | (1) $0.979_{\pm0.001}$ | (6) $0.953_{\pm0.002}$● | (2) $0.978_{\pm0.001}$● | (2) $0.978_{\pm0.001}$● | (5) $0.976_{\pm0.004}$● | (4) $0.977_{\pm0.001}$● |

Table 1: Prediction performance measured by KL and Cosine on logical label enhancement.

model (SABFGS (Geng, 2016) is utilized in this paper). Then, we evaluate the prediction performance of SABFGS on the testing set. Finally, we repeat the above process ten times and report the mean and standard deviation.

**Comparison Algorithms.** We choose two state-of-the-art LE algorithms: CWLD (Fan et al., 2024) and VIB-ILE (Song et al., 2024), and three baseline LE algorithms: GLLE, KMLE, and FCMLE (Xu et al., 2018). The hyperparameter configurations adhere to the respective papers. For CWLD, $\alpha$ is selected among $\{0.1, 0.2, \ldots, 1.0\}$. For VIB-ILE, $\beta$ is selected among $\{10^{-3}, 10^{-2}, \ldots, 10^0\}$, $\gamma$ is selected among $\{2, 20\}$, and $\lambda$ is set to 0.1. For GLLE, $\lambda$ is selected among $\{10^{-2}, 10^{-1}, \ldots, 10^2\}$. For KMLE, $\delta$ is selected among $\{1, 2, 3, 4, 5\}$. For FCMLE, $\beta$ is selected among $\{1, 1.1, 1.2, \ldots, 3\}$. For our method, $\lambda$ and $\beta$ are selected among $\{10^{-1}, 10^0, \ldots, 10^5\}$, $\alpha$ is selected among $\{10^{-2}, 10^{-1}, \ldots, 10^4\}$, the probability distribution $\tilde{p}(\boldsymbol{y}|\boldsymbol{z})$ is modeled by Bernoulli distribution, the parameters of PROM are set as $a = 16, b = 8, t = 3$.

### 4.2.2  Results and Discussions

The prediction performance is shown in Table 1. The rankings of each method are provided in the parentheses before the mean value of performance. Additionally, we use a pairwise two-tailed $t$-test to perform a more thorough comparative analysis. We denote ○/● as the cases that our proposed

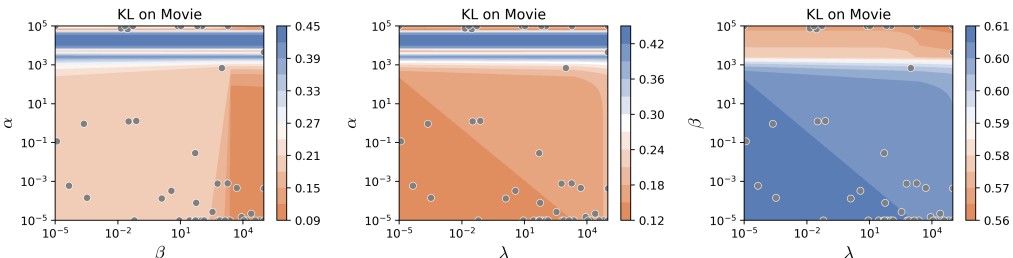

Figure 5: KL-based prediction performance on different values of hyperparameters $\alpha$, $\beta$, and $\lambda$.

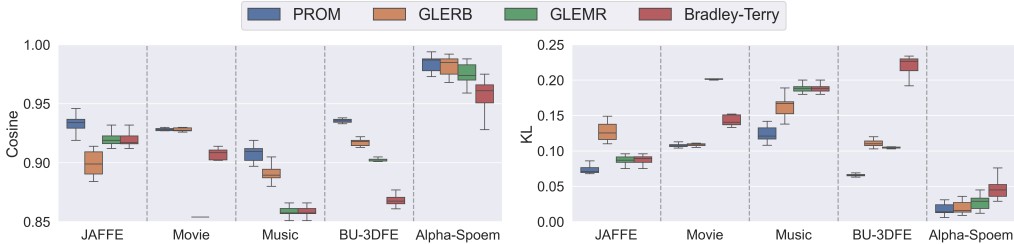

Figure 6: Prediction performance measured by KL and Cosine on label ranking enhancement.

LE-PROM is significantly inferior/superior to the corresponding comparison algorithms. If there is no ○ nor ●, it indicates that there is no significant performance difference between LE-PROM and the corresponding algorithm. Overall, our algorithm achieves competitive prediction performance. Our algorithm achieves an average ranking of 1.09 and wins 138 times, ties 19 times, and loses 3 times, out of 160 statistical significance comparisons.

### 4.2.3 Hyperparameter Analysis

In this subsection, our objective is to demonstrate the effectiveness of each component by varying the values of hyperparameters, i.e. the weights of different components, in LE-PROM. In Figure 5, we show the prediction performance measured by KL on "Movie" dataset with varying values of hyperparameters $\alpha$, $\beta$, and $\lambda$. The experimental results show that both $\alpha$ (the weight of logical label information) and $\beta$ (the weight of label ranking information) have significant impact on the performance of our model. It can be found that an increase in $\alpha$ or $\beta$ usually leads to a performance improvement. However, their values cannot be arbitrarily increased, and when these two hyperparameters reach extremely high values, the performance of the model also decreases. This observation implies that features, logical labels, and label rankings all have an impact on prediction performance, and a proper assignment of weights to these loss terms is critical to model performance. The last two figures in Figure 5 show that $\lambda$ affects the performance to some extent. However, regardless of $\lambda$, optimal performance can typically be achieved by adjusting the $\alpha$ and $\beta$.

### 4.3 Experiments on Label Ranking Enhancement

In this subsection, our objective is to experimentally answer the question of *whether our proposed PROM is superior to current ranking models for the label ranking enhancement task*. The experimental procedure is analogous to Section 4.2.1, with the sole distinction lying in the input label data: one is the logical label, while the other is the label ranking. The training set with label distributions can be directly transformed into the one with label rankings. We choose three comparison algorithms, GLERB (Lu et al., 2023b) and GLEMR (Lu et al., 2023a) are the two state-of-the-art label-enhancement oriented ranking models and Bradley-Terry is a classic ranking model. Since comparison algorithms cannot be directly applied for tie-allowed label rankings, we preprocess the label rankings to accommodate the comparison algorithms. Due to the page limit, we put the details of label preprocessing in Appendix C. From the results in Figure 6, it can be seen that LE-PROM outperforms the comparison algorithms in most cases.

## 5 Conclusion and Limitation Discussion

**Conclusion.** In this paper, we qualitatively and quantitatively study the probabilistic relationship between label distributions and label rankings. We propose assumptions to characterize the probabilistic monotonicity and orderliness of label rankings. Subsequently, we derive a pairwise ranking model PROM that theoretically preserves the probabilistic monotonicity and orderliness. Besides, we propose a generative label enhancement algorithm based on the PROM model to directly learn an LDL mapping on the training instances with multi-label data. The experimental results on extensive real-world datasets demonstrate the superiority of our proposed method.

**Limitation.** Despite the contributions of this paper, it is imperative to acknowledge that the feasible region of PROM exhibits considerable complexity. This complexity may potentially impede the hyperparameter selection. Besides, if users attempt to adaptively learn the parameters $a, b, t$, the feasible region may lead to a difficult optimization problem. Hence, future research efforts will be directed towards simplifying the boundaries of the feasible region, with the aim of enhancing its compatibility with more established optimization methods.

## 6 Acknowledgments

This work was partially supported by the National Natural Science Foundation of China (62176123, 62476130, 62576166), the Natural Science Foundation of Jiangsu Province (BK20242045), the Innovation and Technology Fund (GHP/079/22SZ, ITS/034/23FP), and the UGC/GRF (No. 15211024, 15215421).

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
