# OpenReview forum: "Towards a Pairwise Ranking Model with Orderliness and Monotonicity for Label Enhancement"
_NeurIPS.cc/2025/Conference — NeurIPS 2025 spotlight_

### Official Review · Reviewer_78KF · 2025-06-17

**Clarity:** 3
**Significance:** 3
**Originality:** 3
**Rating:** 5
**Confidence:** 4

**Summary:**

This paper presents PROM (Pairwise Ranking model with Orderliness and Monotonicity) to quantitatively inscribes the probabilistic relationship between the label distribution and the tie-allowed label rankings. The main contributions include formulating the monotonicity and orderliness assumptions of the ranking probabilities, deriving a theoretically-guaranteed ranking model that preserves the monotonicity and orderliness properties, and finally develop LE-PROM, a PROM-grounded generative label enhancement framework. Finally, extensive experiments on 16 datasets are conducted to show the effectiveness of the proposal.

**Questions:**

1. This paper sets $a= 16$, $b= 8$, and $t= 3$. Could authors elaborate more in depth on the potential effects that $a$, $b$ and $t$ would have on the model performance?
2. The feasible domain $\mathcal{F}$ seems restrictive. Could authors provide a preliminary discussion on how to adaptively learn $(a,b,t)$?

**Ethical Concerns:**

["NO or VERY MINOR ethics concerns only"]

**Final Justification:**

The authors have addressed my concerns and thus I maintain my previous rating.

**Limitations:**

yes

**Quality:**

3

**Strengths And Weaknesses:**

Strengths：
This work has several significant strengths. First, the paper qualitatively portrays the probabilistic relationship between label distribution and label ranking by making explicit assumptions (Assumptions 2.1-2.4). These assumptions (i.e., monotonicity, orderliness, translational invariance, and ranking symmetry) are carefully worked out and are highly applicable in real-world scenarios, providing a principled basis for the proposed PROM model. The thorough theoretical analysis (Theorems 2.5-2.8) ensures that the derived PROM model complies with the stated assumptions. The visualization of the theoretical results presented in Figures 2 and 3 also effectively validates the model's adherence to the principles of monotonicity and orderliness. Another major strength of the paper lies in the innovation of the methodology. This paper designs a PROM-grounded generative label enhancement framework, which integrates label ranking, multi-label data, and label distribution learning into a unified probabilistic model. Finally, the effectiveness of the paper's proposal is verified by extensive experiments.

Weaknesses：
This work also has two minor limitations. First, the paper empirically sets the parameters of the PROM (e.g., $a=16, b=8, t=3$) without explaining the motivation of this setting and the sensitivity of the model performance w.r.t. $a$, $b$, and $t$. A brief discussion of the performance sensitivity could be added. Besides, the paper omits some experimental details, such as the choice of optimizer, the learning rate, and so on.

---

> ### Author Rebuttal · Authors · 2025-07-30
>
> Dear Reviewer 78KF,
>
> Thank you for your positive assessment on our work. We sincerely appreciate the time and effort that you have dedicated to evaluating our work. We have carefully considered all the comments and have revised the paper accordingly. Below, we provide a point-by-point response to your suggestions.
>
> ***Question 1: Elaborate more on an in-depth explanation of how the parameters $a$, $b$, and $t$ influence the model's performance.***
>
> The parameters $a$, $b$, and $t$ jointly control the sensitivity of the ranking probabilities. Specifically, $a$ controls the slope of $p(\\ell _ i\\prec\\ell _j)$ and $p(\ell_i\succ\ell_j)$ with respect to the difference of label description degree $z _ i - z _ j$. A larger value of $a$ makes the model more sensitive to the difference of label description degree. The parameter $b$ determines the width of the “approximately the same” ($\\ell _ i\\simeq\\ell _ j$) region. A larger value of $b$ encourages the model to assign equivalence relations more frequently. The parameter $t$ adjusts the baseline probability. A lower value of $t$ increases the model’s tendency to prefer "$\\ell _ i \\prec \\ell _ j$" or "$\\ell _ i\\succ \\ell _ j$" relations when the difference of label description degree is small.
>
> ***Question 2: Provide a preliminary discussion on how to adaptively learn $(a, b, t)$.***
>
> As shown in Equation (4), $\\mathcal F$ imposes structural constraints on the parameters $(a, b, t) $ to ensure that the model satisfies the probability orderliness assumption. While these constraints are theoretically necessary, they do not impose a fundamental limitation on the training process. During optimization, one can apply projected gradient descent to update $(a, b, t)$ while ensuring that the parameters remain within $\\mathcal F$. Alternatively, a reparameterization approach can be used to express $(a, b, t)$ in terms of unconstrained variables through differentiable transformations, allowing implicit enforcement of the constraints. We will further explore adaptive strategies to better reconcile theoretical feasibility with practical flexibility in training.

---

### Official Review · Reviewer_jHPx · 2025-06-25

**Clarity:** 3
**Significance:** 4
**Originality:** 3
**Rating:** 5
**Confidence:** 4

**Summary:**

The authors theoretically investigate a number of properties of the mapping from the latent label distribution to the label rankings: probabilistic orderliness and probabilistic monotonicity. The authors rigorously define these two properties and derive the conditions under which the proposed distribution-conditioned label ranking model satisfies both properties. In addition, the authors propose a generalized label enhancement model that can be directly utilized as a label distribution predictor based on label ranking. Finally, the authors validate their proposal on several real-world datasets.

**Questions:**

1. Equation (4) seems difficult to understand; could it be transformed into an easily understandable form?

2. In Equation (2), why should the probability distribution of $\ell_i=\ell_j$ be modeled by the square of the distance between $z_i$ and $z_j$, while the other two ranking relationships are modeled by linear functions of the distance.

**Ethical Concerns:**

["NO or VERY MINOR ethics concerns only"]

**Final Justification:**

The author addressed my concerns regarding Equations 2 and 4 in the paper. Additionally, considering the feedback from other reviewers, I have decided to keep my score unchanged.

**Limitations:**

yes

**Quality:**

3

**Strengths And Weaknesses:**

Strengths：
1. This paper delves into a meaningful problem of what properties the relationship between the label ranking and the underlying label distribution possesses, and how a label ranking model can be constructed to maintain these properties.

2. This paper theoretically and methodologically studies these two problems, and the theoretical results and the proposed PROM model potentially contribute many new ideas to the field of weakly-supervised machine learning.

3. The authors propose a label distribution generation model based on PROM, which achieves outstanding performance on several real-world datasets.

Weaknesses:
1. There are some undefined terms in this paper, such as description degree in line 80-84.

2. The writing of this paper can be further improved, such as splitting some complex sentences and embellishing the expressions.

---

> ### Author Rebuttal · Authors · 2025-07-29
>
> Dear Reviewer jHPx,
>
> Thank you for your positive assessment on our work. We sincerely appreciate the time and effort that you have dedicated to evaluating our work. We have carefully considered all the comments and have revised the paper accordingly. Below, we provide a point-by-point response to your suggestions.
>
> ***Question 1: Equation (4) seems difficult to understand; could it be transformed into an easily understandable form?***
>
> The function $\\mathcal F$ is introduced to ensure that the predicted probabilities satisfy the required ordering constraints. The Equation (4) can be decomposed as follows.
>
> **Basic Conditions**
>
> - $b > 0$
> - $t > 0$
> - $a > |b - t|$
>
> **Additional Conditions (Branching Based on the Relationship Between $a$ and $b$)**
>
> - **If $a \geq 2b$**:
>   No additional conditions are required (automatically satisfied).
>
> - **If $a < 2b$**:
>   Further check the following subconditions:
>
>   - **If $a^2 \geq 4b^2 - 2b$**:
>     An additional constraint is required:
>     $t< b + a - \log \frac{2b - a}{2a}$.
>
>   - **If $a^2 < 4b^2 - 2b$**:
>     Then the following tighter constraint is needed:
>     $t< \frac{2a \sqrt{a^2 + 2b} + a^2}{4b} - \log \frac{\sqrt{a^2 + 2b} - a}{2a} + \frac{1}{2}$.
>
> ***Question 2: In Equation (2), why should the probability distribution of $\\ell _ i=\\ell _ j$ be modeled by the square of the distance between $\\boldsymbol z_i$ and $\\boldsymbol z_j$, while the other two ranking relationships are modeled by linear functions of the distance?***
>
> Considering the symmetry of the probability distribution of $\\ell _ i=\\ell _ j$ with respect to $\\boldsymbol z _i - \\boldsymbol z _j$, we initially employed the absolute difference between $\\boldsymbol z_i$ and $\\boldsymbol z_j$ in our formulation, which maintains the same power-law relationship with respect to $\\boldsymbol z _i- \\boldsymbol z _j$ as observed in the probabilities of both $\\ell _ i\\prec \\ell _ j$ and $\\ell _ i\\succ\\ell _ j$. However, this method gives rise to a very complex analytical form of PROM, which may hinder the model training process. Therefore, we model the probability distribution of $\\ell _ i=\\ell _ j$ by the square of the distance between $\\boldsymbol z_i$ and $\\boldsymbol z_j$.

---

> > ### Comment · Reviewer_jHPx · 2025-08-07
> >
> > Thank you for the clarifications. My concerns have been addressed. After considering the feedback from other reviewers, I will maintain my positive recommendation.

---

### Official Review · Reviewer_QeX9 · 2025-06-27

**Clarity:** 2
**Significance:** 3
**Originality:** 3
**Rating:** 4
**Confidence:** 4

**Summary:**

The paper introduces the PROM method to handle the ranking issue of the probability between label distribution and tie-allowed label rankings; a label enhance approach designed to capture ranking relation. Based on PROM, LE-PROM is designed for logical label enhancement. The authors validate the effectiveness of LE-PROM through extensive experiments on sixteen benchmark datasets, comparing it with five existing label enhance algorithms.

**Questions:**

1. Please describe the addressed issue of this paper in a more concise manner in the abstract.
2. The motivation of this paper is not clear. The problem of label ranking has been solved by many researchers. Authors should indicate the limitations of the existing works and proposed appropriate solutions to overcome them.
3. In the section of related work, the advantages and disadvantages of the existing works should be analyzed and summarized. In addition, the differences between the proposed method and the existing works should be clearly indicated.
4. For the proposed method, each step is not well described.
5. Most of the paper contains many shorthand notations and abstract level information. Each and every term of the equation should be clearly explained.
6. If you want to show you work in more detail, please upload both code and data to github, and provide the link in the paper.

**Ethical Concerns:**

["NO or VERY MINOR ethics concerns only"]

**Final Justification:**

Rating: 4: Borderline accept: Technically solid paper where reasons to accept outweigh reasons to reject, e.g., limited evaluation. Please use sparingly.

The authors have answered all my concerned questions. However, there are some minor issues that need to be addressed, such as the incomplete information of reference. The presentation can be further improved. I maintain a borderline accept.

**Limitations:**

Yes.

**Paper Formatting Concerns:**

None.

**Quality:**

3

**Strengths And Weaknesses:**

Strengths
1. PROM introduces an approach to consider label ranking correlation and prove the monotonicity and orderliness, making it a contribution to the field of LDL.
2. The paper conducts experiment on sixteen benchmark datasets, employing four evaluation metrics to demonstrate the label enhance effectiveness of LE-PROM. Additionally, LE-PROM is compared to five state of-the-art label enhance algorithms, highlighting its competitive performance.

Weaknesses
1. The motivation of this paper is not clear. The issue of label ranking has been solved by many researchers. Authors should indicate the limitations of the existing works and proposed appropriate solutions to overcome them.
2. The analysis and discuss of related work are little. In the section of related work, the advantages and disadvantages of the existing works should be analyzed and summarized. In addition, the differences between the proposed method and the existing works should be clearly indicated.
3. The presentation is not well. The description of the proposed method and the meanings of notations are ambiguous.

---

> ### Author Rebuttal · Authors · 2025-07-29
>
> Dear Reviewer QeX9,
>
> Thank you for your constructive comments on our paper. We sincerely appreciate the time and effort that you have dedicated to evaluating our work. We have carefully considered all the comments and have revised the paper accordingly. Below, we provide a point-by-point response to your suggestions.
>
> ***Question 1: Describe the addressed issue in a more concise manner in the abstract.***
>
> We carefully revise the abstract to better highlight the addressed issue of this paper. Specifically, numerous works have demonstrated that the label ranking is significantly beneficial to label enhancement. However, existing works either exhibit research gaps in quantitatively and qualitatively characterizing the probabilistic relationship between the label distribution and their corresponding TALR (tie-allowed label rankings), or cannot account for the cases where labels are equally important to an instance. We will revised the abstract according to your comments in the final version.
>
> ***Q2: Authors should indicate the limitations of the existing works and proposed appropriate solutions to overcome them.***
>
> Our work aims to address a fundamental yet understudied problem in label enhancement, i.e., establishing both qualitative and quantitative relationships between label distributions and TALR. Specifically, we investigate: (1) how changes in label distributions affect trends in TALR, and (2) how the magnitudes of label distributions influence probabilistic relationships in TALR. These investigations provide crucial guidance for designing probabilistic generative processes from label distributions to rankings and developing loss functions that measure inconsistencies between label distributions and TALR.
>
> **Current Limitations in Ranking Probability Models.** While existing ranking probability models (e.g., Bradley-Terry and Plackett-Luce) can quantitatively describe relationships between object logits and strict rankings, they suffer from a critical limitation that they only model strict ordering relations ($\succ$ or $\prec$), i.e., they cannot handle tie cases where labels describe instances equally. Although [1] extended Bradley-Terry model to TALR by assumes ties occur when $z_i = z_j$, i.e., $p(\ell_i\prec\ell_j)=p(\ell_i\succ\ell_j)=0.5$, it fails to account for real-world uncertainties in tie relation. However, as claimed by [2], in real-world situations the labels with description degrees smaller than a certain distance can be treated as tied labels.
>
> **Current Limitations in Existing Approaches for Tie Modeling.** Several works have attempted to address tie cases. For example, [2] introduces margin concepts for uncertain ties, [3] assigns equal weights to tied labels in loss functions. [4] utilizes virtual labels and bucket orders to model uncertainty. However, these methods share a common shortcoming: They merely assume all distributions satisfying ranking constraints are equally probable, without deeper investigation into the fundamental qualitative and quantitative relationships.
>
> **Our Novel Contributions**
> Therefore, we attempt to investigate the qualitative and quantitative relationships between TALR and label distribution within the task framework of label enhancement. The qualitative study focuses on how changes in label distribution may alter trends in TALR, as well as how the magnitude of label distribution affects probabilistic ordinal relationships in TALR. The quantitative study aims to formalize the findings from qualitative research from a probabilistic perspective.
>
> [1] X. Jia, X. Shen, W. Li, Y. Lu and J. Zhu, "Label Distribution Learning by Maintaining Label Ranking Relation," IEEE Transactions on Knowledge and Data Engineering, vol. 35, no. 2, pp. 1695-1707.
>
> [2] Y. Lu, W. Li, H. Li and X. Jia, "Predicting Label Distribution From Tie-Allowed Multi-Label Ranking," IEEE Transactions on Pattern Analysis and Machine Intelligence, vol. 45, no. 12, pp. 15364-15379.
>
> [3] X. Jia, T. Qin, Y. Lu and W. Li, "Adaptive Weighted Ranking-Oriented Label Distribution Learning," IEEE Transactions on Neural Networks and Learning Systems, vol. 35, no. 8, pp. 11302-11316.
>
> [4] X. Geng, R. Zheng, J. Lv and Y. Zhang, "Multilabel Ranking With Inconsistent Rankers," IEEE Transactions on Pattern Analysis and Machine Intelligence, vol. 44, no. 9, pp. 5211-5224.
>
> ***Question 3: The advantages and disadvantages of the existing works should be analyzed and summarized. The differences between the proposal and the existing works should be clarified.***
>
> **Advantages and disadvantages of existing works.** Due to the character limits, we briefly discuss the pros and cons of existing works. In terms of LE (label enhancement), existing works can be divided into ranking-based LE and ranking-free LE. Ranking serving as the key information underlying the label distribution is essential for LE. Hence, a portion of LE research recently explores LE from the perspective of label ranking, such as LEPNLR, GLEMR, GLERB, DRAM. The detailed description of these methods can be found in the main text. These methods focus on how to design a ranking-preserved function to regularize the space of label distribution. However, these methods exhibit deficiencies in qualitatively and quantitatively representing the probabilistic relationships between the label distribution and the TALR. Specifically, they either assume that all TALR-preserving label distributions are equiprobable (i.e., they ignore the probabilistic differences among distinct label distributions that satisfy the same TALR, for instance, the probability of $\ell_i\prec \ell_j$ is certainly higher when $z_i−z_j=0.9$ compared to when $z_i−z_j=0.1$, or they fail to account for tie cases. In terms of ranking model, most existing approaches either fail to model scenarios where labels are equally important for instances (e.g., Bradley-Terry and Plackett-Luce), or cannot represent TALR as a probability distribution conditioned on the label distribution (e.g., [Modifying Bradley–Terry and other ranking models to allow ties]).
>
> **Differences between the proposed method and the existing works.** Our paper establishes four key properties between TALR and label distributions: *Probability Monotonicity*, *Probability Orderliness*, *Translational Invariance*, and *Ranking Symmetry*. These properties are rigorously integrated into a ranking probability model, thereby bridging the gap in existing literature regarding both qualitative and quantitative relationships between label distributions and TALR.
>
> ***Question 4: For the proposed method, each step is not well described.***
>
> In summary, we propose a probability distribution named PROM and a generative label enhancement framework called LE-PROM based on PROM. The probability mass function of PROM is given by Equation (3) in the paper, with the value range of each parameter constrained by Equation (4). LE-PROM consists of a probabilistic generative process for observed variables (features, readily available labels, TALR) and a posterior inference process for latent variables (label distribution). We assume that the label distribution, as latent variables, generates all observed variables. Specifically, the feature vector is sampled from a Gaussian distribution conditioned on a nonlinear mapping of the label distribution, the readily available labels are  generated from a task-specific distribution, and TALR is generated from the PROM distribution conditioned on the label distribution. For inferring the latent label distribution, we adopt Auto-Encoding Variational Bayes (AEVB), which approximates the true posterior using a parameterized variational distribution. AEVB transforms latent variable inference into an optimization problem, where the objective combines a reconstruction term for observed variables (which can be estimated via Monte Carlo sampling), and a KL divergence term between the variational posterior and prior distributions (which can be computed analytically). To further obtain a feature-conditioned label distribution predictor, we introduce an additional loss term that minimizes the KL divergence between the predictor’s output and the variational posterior of the label distribution. Finally, the feature-conditioned label distribution predictor can be learned by Equation (8) in the paper.
>
> ***Question 5: Most of the paper contains many shorthand notations and abstract level information. Each term should be clearly explained.***
>
> We carefully checked all formulas and summarized the meanings of the commonly used shorthand terms and mathematical symbols as follows:
> |Symbol|Meaning|
> |-|-|
> |LDL|label distribution learning|
> |LE|label enhancement|
> |ELBO|evidence lower bound|
> |$\mathcal{X}^D$|the $D$-dimensional feature space|
> |$\Delta^M$|the $M$-dimensional label distribution space|
> |$\ell_m$|the $m$-th label|
> |$v_m$| the description degree of label $\ell_m$|
> |$\mathcal{Y}$| the label space containing all possible labels $\{\ell_1, \ell_2,...,\ell_M\}$|
> |$N$|the number of instances in the training dataset|
> |$\boldsymbol x_n$| the feature vector of the $n$-th instance|
> |$\boldsymbol y_n$  | the vector of the easily available label values of the $n$-th instance|
> |$\ell_i \prec \ell_j$, $\ell_i\succ\ell_j$, $\ell_i\simeq\ell_j$ | label $i$ describes the instance to a higher, lower, and approximately the same degree compared to label $j$|
> |$\xi_{ij}$  | the ranking relation between labels $\ell_i$ and $\ell_j$ |
> |$z_i$  | the description degrees of labels $\ell_i$|
> |$\underline{\phi}(z_i, z_j)$ | probability that $\ell_i \succ \ell_j$|
> |$\tilde{\phi}(z_i, z_j)$| probability that $\ell_i \simeq \ell_j$|
> |$\overline{\phi}(z_i, z_j)$ | probability that $\ell_i \prec \ell_j$|
>
> ***Question 6: Upload both code to github, provide links in the paper.***
>
> We have now uploaded the code and data to GitHub and will include the link in the revised paper. We appreciate your feedback and hope this improves the reproducibility of our work.

---

> > ### Comment · Reviewer_QeX9 · 2025-08-05
> >
> > Thanks to the authors for answering my questions and taking the time to revise the paper. I am willing to update my score. However, there are some minor issues that need to be addressed, such as the incomplete information of reference. Please do include the revisions as I have yet to see it show up in the pdf linked above.

---

> > > ### Author Response · Authors · 2025-08-05
> > >
> > > Dear Reviewer QeX9,
> > >
> > > Thank you for your positive feedback. We greatly appreciate your willingness to update your score. Regarding the minor issues you mentioned, particularly the incomplete reference information, we have already made the aforementioned revisions in our manuscript. However, we currently don't have the option to upload the revised PDF at this stage of the review process. We will submit the updated version with all revisions included during the next available submission window.

---

### Official Review · Reviewer_M8yu · 2025-06-29

**Clarity:** 3
**Significance:** 3
**Originality:** 3
**Rating:** 5
**Confidence:** 5

**Summary:**

In this paper, the authors propose PROM, a pairwise ranking model with orderliness and monotonicity, to capture the probabilistic relationship between label distributions and label rankings.
The proposed method aims to address the following two limitations of the current label enhancement (LE) process:
1. Ignorance of the ranking relation in ties.
2. Ignorance of the probabilistic relationships between label distributions and label rankings.
Since existing classic ranking models, such as the Bradley–Terry and Plackett–Luce models, cannot explicitly model tie relations, the authors conduct a qualitative analysis of the generation process of label rankings from label distributions and design a parameterized probability mass function for the ranking model.
Based on these, they propose a generative label enhancement algorithm called LE-PROM and validate their approach through extensive experiments on real-world datasets.

**Questions:**

1. In ELBO terms, only the approximation of  \mathbb{E}_{q(u|x,y,\xi)}[\log p(x,y,\xi|u)]  is given. What about the detail calculation of \mathcal{D}_{KL}(q||p(u))  and  \mathcal{D}_{KL}(\hat{p}||q) ? They are not clearly described.

**Ethical Concerns:**

["NO or VERY MINOR ethics concerns only"]

**Final Justification:**

All my concerns have been resolved, and I am now inclined to recommend acceptance of the paper.

**Limitations:**

yes

**Paper Formatting Concerns:**

The paper formatting appears to meet conference standards with no major concerns.

**Quality:**

3

**Strengths And Weaknesses:**

Strength:

1. (Idea and Method) The proposed method adequately explores the relationship between label ranking and label distribution/description, with particular attention to the analysis of tie rankings. Specifically, the discussion on the relationship between label ranking and label distribution is thorough.
2. (Theory) The theoretical foundation of the proposed method is rigorous. The probabilistic modeling of label ranking is reasonable under the stated assumptions.
3. (Experiments) The experimental design is sufficient and valid. The authors conduct experiments on 16 real-world datasets, comparing PROM with several state-of-the-art methods. The datasets are preprocessed using min-max normalization, and the experiments are repeated 10 times to ensure statistical reliability. KL divergence and cosine similarity are used as evaluation metrics, which are appropriate for assessing the quality of label distributions.

Weakness:

1. There may be a critical misdescription in the formulation of the proposed method. Specifically, the interpretation appears to be reversed. The key issue lies in why a larger value of $z_i-z_j$ implies a larger probability $p(\ell_{i}\prec \ell_{j}|z_{i},z_{j})$ in Assumption 2.1 and 2.2. This seems counterintuitive. Should it instead be that a larger $z_{i}-z_{j}$ implies a larger probability  $p(\ell_{i}\succ \ell_{j}|z_{i},z_{j})$?
2. The assumption underlying the generative model may be questionable. In the generative process, the authors assume that $p(x,y,u)=p(x|u)\cdot p(y|u)\cdot p(u)$ (Here the variable $\xi$ is ignored).  This implies that, given $u$ is known, the variable $x,y$ is independent, i.e., $p(x,y|u)=p(x|u)\cdot p(y|u)$, meaning that the logical label $y$ is independent of $x$ given $u$. This is clearly unreasonable, as many multi-label learning methods have demonstrated that there is a strong dependency between between $x$ and $y$. The questionable nature of this assumption casts doubt on the reliability of the experimental results. Although simplifying assumptions are common, this particular assumption appears overly strong.

---

> ### Author Rebuttal · Authors · 2025-07-29
>
> Dear Reviewer M8yu,
>
> Thank you for your positive assessment on our work. We sincerely appreciate the time and effort that you have dedicated to evaluating our work. We have carefully considered all the comments and have revised the paper accordingly. Below, we provide a point-by-point response to your suggestions.
>
> ***Question 1: Only provides the approximation for $\\mathbb E _ \{q(u | x,y,\\xi)\}[\\log p(x,y,\\xi | u)]$, while the computations of $\\mathcal D _ \{KL\}(q | p(u))$ and $\\mathcal D _ \{KL\} (\\hat p | q)$ are not clearly explained.***
>
> In terms of $\\mathbb E _ \{q(u|x,y,\\xi)\}[\\log p(x,y,\\xi|u)]$, since the joint distribution $p(x, y, \\xi \\mid u)$ involves the generation process of $x$, $y$, and $\\xi$, its form can be quite complex, making the expectation with respect to $u$ intractable in closed form. Therefore, we employ Monte Carlo approximation to estimate the expectation term $\\mathbb\{E\} _ \{q(u|x, y,\\xi)\}[\\log p(x, y, \\xi \\mid u)]$.
>
> In terms of $\\mathcal D _ \{KL\}(q | p(u))$ and $\\mathcal D _ \{KL\}(\\hat p | q)$, the KL divergence terms involve Gaussian distributions, for which closed-form solutions can be obtained analytically. Therefore, we did not repeat the derivation in the paper for the sake of page limits. Nevertheless, we acknowledge your concern regarding the lack of explicit descriptions. We provide the detailed formulations of the two KL divergence terms as follows.
> For the KL divergence between the variational posterior $q(u|x, y, \\xi) = \\mathcal N (\\mu _ q, \\Sigma _ q)$ and the prior $p(u) = \\mathcal N (\\mu _ p, \\Sigma _ p)$, the closed-form is:
>
> $$
> \\mathcal D _ \{\\mathrm KL\}(q(u|x, y, \xi) | p(u)) = \\frac{1}{2} \\sum _ \{i=1\}^d \\left[ \\log \\frac\{(\\Sigma _ p)_i\}\{(\\Sigma _ q)_i\} + \\frac\{(\\Sigma _ q)_i + (\\mu _ q)_i^2 - 2(\\mu _ q)_i(\\mu _ p)_i + (\\mu _ p)_i^2\}\{(\\Sigma _ p)_i\} - 1 \right]
> $$
>
> Similarly, for the KL divergence between the enhanced prior $\\hat p (u) = \\mathcal N (\\mu _ \{\\hat p\}, \\Sigma _ \{\\hat p\})$ and the posterior $q(u|x, y, \\xi)$, the expression is:
>
> $$
> \\mathcal D _ \{ \\mathrm KL \} (\\hat p (u) | q(u)) = \\frac12 \\sum _ \{i=1\}^d \\left[ \\log \\frac\{ (\\Sigma_q) _ i \} \{(\\Sigma_\{ \\hat p \}) _ i\} + \frac\{ (\\Sigma_\{ \\hat p \}) _ i + (\\mu_\{\\hat p \}) _ i^2 - 2(\\mu_\{\\hat p \}) _ i (\\mu _ q)_i + (\\mu _ q)_i^2 \}\{ (\\Sigma _ q)_i \} - 1
> \\right]
> $$

---

> > ### Comment · Reviewer_M8yu · 2025-08-06
> >
> > Thank you for your thoughtful response. My concerns have been addressed, and I will raise my score accordingly.

---

### Decision · Program_Chairs · 2025-09-17

**Decision:**

Accept (spotlight)

**Comment:**

This paper focuses on label enhancement, which infers label distributions from multi-label data, and highlights the limitations of existing methods in modeling probabilistic relationships between distributions and tie-allowed label rankings. To address this, it proposes PROM, a pairwise ranking model with orderliness and monotonicity assumptions, derives corresponding mass functions with theoretical guarantees, and develops a generative label enhancement algorithm based on PROM. The reviewers generally provided positive feedback, recognizing both the methodological contributions and the theoretical underpinnings. Overall, the paper is recommended for acceptance.